# Biosynthetic tailoring of existing ascaroside pheromones alters their biological function in *C. elegans*

Yue Zhou†, Yuting Wang†, Xinxing Zhang, Subhradeep Bhar, Rachel A Jones Lipinski, Jungsoo Han, Likui Feng, Rebecca A Butcher*

Department of Chemistry, University of Florida, Gainesville, United States

**Abstract** *Caenorhabditis elegans* produces ascaroside pheromones to control its development and behavior. Even minor structural differences in the ascarosides have dramatic consequences for their biological activities. Here, we identify a mechanism that enables *C. elegans* to dynamically tailor the fatty-acid side chains of the indole-3-carbonyl (IC)-modified ascarosides it has produced. In response to starvation, *C. elegans* uses the peroxisomal acyl-CoA synthetase ACS-7 to activate the side chains of medium-chain IC-ascarosides for β-oxidation involving the acyl-CoA oxidases ACOX-1.1 and ACOX-3. This pathway rapidly converts a favorable ascaroside pheromone that induces aggregation to an unfavorable one that induces the stress-resistant dauer larval stage. Thus, the pathway allows the worm to respond to changing environmental conditions and alter its chemical message without having to synthesize new ascarosides de novo. We establish a new model for biosynthesis of the IC-ascarosides in which side-chain β-oxidation is critical for controlling the type of IC-ascarosides produced.
DOI: https://doi.org/10.7554/eLife.33286.001

*For correspondence:
butcher@chem.ufl.edu

†These authors contributed equally to this work

Competing interests: The authors declare that no competing interests exist.

## Introduction

The nematode *C. elegans* secretes ascarosides as pheromones to induce development of the dauer larval stage at high population densities, as well as to control various behaviors, including male attraction to hermaphrodites, hermaphrodite attraction to males, avoidance, foraging behavior, and adult aggregation (*Cassada and Russell, 1975*; *Golden and Riddle, 1982*; *Butcher et al., 2007*; *Butcher et al., 2008*; *Srinivasan et al., 2008*; *Butcher et al., 2009a*; *Pungaliya et al., 2009*; *Izrayelit et al., 2012*; *Srinivasan et al., 2012*; *Artyukhin et al., 2013*; *Greene et al., 2016*). The ascarosides are derivatives of the 3,6-dideoxysugar ascarylose that can be named according to their structural features, including the number of carbons (C#) in their fatty acid-derived side chains (*Butcher, 2017*). These side chains are attached to the ascarylose sugar at their penultimate ($\omega-1$) or terminal ($\omega$) carbon and are often unsaturated ($\Delta$) at the $\alpha$-$\beta$ position. The ascarosides can be modified with various head groups at the 4′-position of the ascarylose sugar, such as the IC, the octopamine succinyl (OS), the 4-hydroxybenzoyl (HB), and the (*E*)-2-methyl-2-butenoyl (MB) groups. They can also be modified with various terminus groups at the end of their fatty-acid side chain. These head and terminus modifications can have important consequences for the biological activity of the ascarosides. For example, while unmodified, simple ascarosides are often negative signals that induce avoidance in *C. elegans*, ascarosides that are modified with an IC head group, such as IC-asc-$\Delta$C9 (icas#3), are favorable signals that induce aggregation at picomolar concentrations (*Srinivasan et al., 2012*; *von Reuss et al., 2012*). However, the consequences of the IC group and other modifications can be quite nuanced. For example, unlike IC-ascarosides with medium-length (C9) side chains, which do not induce dauer, an IC-ascaroside with a short (C5) side chain, IC-asc-C5 (icas#9), is one of the five primary components of the dauer pheromone and induces the dauer larval

**eLife digest** Small roundworms such as *Caenorhabditis elegans* release chemical signals called ascarosides in order to communicate with other worms of the same species. Using the ascarosides, the worm can tell its friends, for example, how crowded the neighborhood is and whether there is enough food. The ascarosides thus help the worms in the population decide whether the neighborhood is good – meaning they should hang around, eat, and make babies – or whether the neighborhood is bad. If so, the worms should develop into a larval stage specialized for dispersal that will allow them to find a better neighborhood.

Roundworms make the ascarosides by attaching a long chemical 'side chain' to an ascarylose sugar. Further chemical modifications allow the worms to produce different signals. In general, to signal a good neighborhood, worms attach a structure called an indole group to the ascarosides. To signal a bad neighborhood, worms make the side chain very short. But how does a worm control which ascarosides it makes?

Zhou, Wang et al. now show that *C. elegans* can change the meaning of its chemical message by modifying the ascarosides that it has already produced instead of making new ones from scratch. Specifically, as their neighborhood runs out of food, *C. elegans* can use an enzyme called ACS-7 to initiate the shortening of the side chains of indole-ascarosides. The worm can thus change a favorable ascaroside signal that causes the worms to group together into an unfavorable ascaroside signal that causes the worms to enter their dispersal stage.

Although Zhou, Wang et al. have focused on chemical communication in *C. elegans*, the findings could easily apply to the many other species of roundworm that produce ascarosides. Knowing how worms communicate will help us to understand how worms respond to their environment. This knowledge could potentially be used to interfere with the lifecycles and survival of parasitic worm species that harm health and crops.

DOI: https://doi.org/10.7554/eLife.33286.002

stage when it accumulates at low nanomolar concentrations (*Butcher et al., 2009a*). Thus, IC-ascarosides have a dual nature to their activity as they are generally favorable signals that induce aggregation on food, but can also indicate unfavorable conditions and induce dauer when they have short side chains and accumulate to higher concentrations.

Peroxisomal β-oxidation is central to biosynthesis of the ascarosides. β-oxidation cycles shorten the side chains of long-chain ascaroside precursors by two carbons per cycle to produce short- and medium-chain ascaroside pheromones (*Golden and Riddle, 1985*; *Wanders and Waterham, 2006*; *Butcher et al., 2009b*; *Joo et al., 2009*; *Joo et al., 2010*; *von Reuss et al., 2012*; *Zhang et al., 2015*; *Zhang et al., 2018*). The first step in these β-oxidation cycles is catalyzed by an acyl-CoA oxidase, which installs a double bond at the α-β position of its substrate in a reaction that uses an FAD cofactor and produces $H_2O_2$. Specific acyl-CoA oxidases process ascaroside precursors with specific side-chain lengths and thus help to control which pheromones are produced (*Joo et al., 2010*; *von Reuss et al., 2012*; *Zhang et al., 2015*; *Zhang et al., 2018*). The *C. elegans* genome encodes seven acyl-CoA oxidases, five of which have been implicated in ascaroside production (*Zhang et al., 2018*). To reflect their similarity to mammalian ACOX-1, the acyl-CoA oxidases that were previously referred to as ACOX-1, –2, –3, –4, and –5 (*von Reuss et al., 2012*; *Zhang et al., 2015*; *2016*) have been renamed ACOX-1.1, –1.2, –1.3, –1.4, and –1.5, respectively, and an additional acyl-CoA oxidase (F59F4.1) has been named ACOX-1.6. To reflect its similarity to mammalian ACOX-3, the acyl-CoA oxidase that was previously referred to as ACOX-6 (*Zhang et al., 2016*) has been renamed ACOX-3. The remaining three steps in each β-oxidation cycle are catalyzed by an enoyl-CoA hydratase (MAOC-1), an (*R*)-3-hydroxyacyl-CoA dehydrogenase (DHS-28), and a 3-ketoacyl-CoA thiolase (DAF-22) (*Butcher et al., 2009b*; *Joo et al., 2009*; *Joo et al., 2010*; *von Reuss et al., 2012*; *Zhang et al., 2015*). Our group has shown that (ω−1)-ascarosides and ω-ascarosides are biosynthesized by two parallel β-oxidation pathways, the former pathway involving ACOX-1.1, ACOX-1.3, ACOX-1.4, and ACOX-3 and the latter pathway involving ACOX-1.1 and ACOX-1.2 (*Zhang et al., 2015*; *Zhang et al., 2018*). In general, extensive β-oxidation of

ascaroside precursors by these pathways to make ascarosides with very short side chains generates some of the most potent dauer pheromones, such as asc-ωC3 (ascr#5).

Very little is known about how various head groups, such as the IC, OS, HB, and MB groups, become attached to the ascarosides and how this attachment process is coordinated with the β-oxidation process that shortens the ascaroside side chains. Attachment of the IC head group to the 4'-position of the ascarylose is thought to occur late in the biosynthetic pathway after β-oxidation of the side chain has occurred (*von Reuss et al., 2012*). The exogenous addition of a synthetic ascaroside with a nine-carbon side chain, asc-C9 (ascr#10), to *daf-22* worms (which are defective in β-oxidation and thus cannot make short- and medium-chain ascarosides, but which presumably can still biosynthesize the IC group and attach it) leads to production of the corresponding IC-ascaroside, IC-asc-C9 (icas#10) (*von Reuss et al., 2012*). This result suggests that attachment of the IC group occurs after side-chain β-oxidation, at least in the biosynthetic pathway to IC-asc-C9. The acyl-CoA synthetase gene, *acs-7*, has been shown to be required for biosynthesis of the short-chain IC- and OS-modified ascarosides, IC-asc-C5 and OS-asc-C5 (osas#9), but not for medium-chain IC- and OS-ascarosides, such as IC-asc-C9 and OS-asc-C9 (osas#10) (*Panda et al., 2017*). Panda et al. proposed that ACS-7 activates indole-3-carboxylic acid (ICA) as the corresponding CoA-thioester (IC-CoA) and is involved in the attachment of the IC group specifically to the short-chain ascaroside asc-C5 (ascr#9) to make IC-asc-C5 (*Figure 1A*). However, while Panda et al. were able to show that ACS-7 activates ICA as an AMP-ester (IC-AMP), they could not show that ACS-7 catalyzes the formation of IC-CoA or attachment of the IC group to an ascaroside (*Figure 1A*). Their interpretation of these data was that some other protein or factor was required for the full activity of ACS-7. Panda et al. showed that mutant worms lacking lysosome-related organelles (LROs) did not make 4'-modified ascarosides, and they reported that ACS-7 was expressed in the lysosome (*Panda et al., 2017*). Thus, they concluded that ACS-7 was functioning inside the lysosomes in IC-ascaroside biosynthesis.

Here, we show that the role of the acyl-CoA synthetase ACS-7 is not to attach the IC group to ascarosides as proposed by Panda et al., but to activate the side chains of medium-chain IC-ascarosides for shortening through β-oxidation (*Figure 1B*). Thus, in contrast to the previously proposed model, we show that β-oxidation of the side chain occurs *after* attachment of the IC head group in the biosynthesis of the short-chain IC-ascarosides. Our data show that under favorable, growth-promoting conditions, the IC group is preferentially attached to ascarosides with 8–11-carbon side chains to form aggregation pheromones. However, if conditions decline, such as during starvation, the side chains of these IC-ascarosides can be activated by ACS-7 and then shortened through peroxisomal β-oxidation cycles involving ACOX-1.1 and ACOX-3 to make the potent dauer pheromone IC-asc-C5. Consistent with its function in activating ascarosides for peroxisomal β-oxidation, ACS-7 is localized to the peroxisome, rather than the lysosome, as previously reported by Panda et al. Our results uncover the mechanism by which *C. elegans* responds to declining environmental conditions and converts aggregation-inducing, medium-chain IC-ascarosides to dauer-inducing, short-chain IC-ascarosides. This mechanism is likely also used by *C. elegans* in the biosynthesis of short-chain OS-ascarosides, which induce nematode dispersal under unfavorable conditions. Using this mechanism, *C. elegans* can efficiently alter the function of existing ascarosides simply by tailoring the length of the side chain, providing a novel strategy to rapidly modulate chemical signaling in response to environmental conditions.

## Results

### Attachment of the IC group during ascaroside biosynthesis

Previously, it has been shown that *daf-22* worms, which cannot make any short- or medium-chain ascarosides, can attach the IC group to exogenously provided synthetic asc-C9 to produce IC-asc-C9 (structures shown in *Figure 1A,B* and *Figure 2A*) (*von Reuss et al., 2012*). It was also shown that *daf-22* worms can attach the IC group to exogenously provided asc-ΔC9 (ascr#3) to produce IC-asc-ΔC9 (*von Reuss et al., 2012*). To determine whether *C. elegans* could attach the IC group to other ascarosides, we cultured *daf-22* worms with ascarosides of various side-chain lengths (*Figure 2A*). These data suggest that the unknown enzyme that attaches the IC group specifically prefers ascarosides with side chains of 8–11 carbons in length (*Figure 2B*). Recently, Panda et al. indicated that they were also able to detect the conversion by *daf-22* worms of asc-C7 (ascr#1) to IC-asc-C7

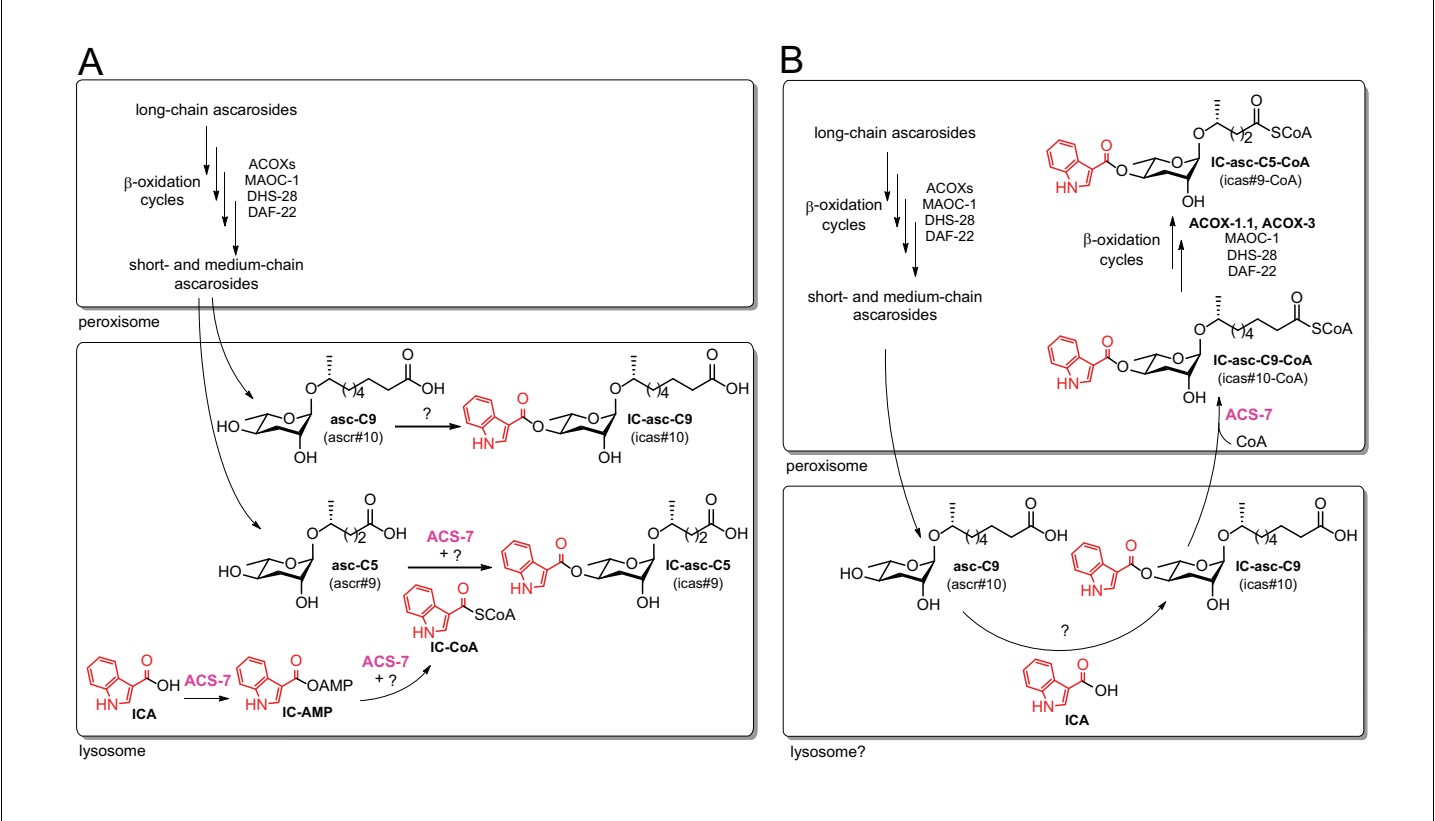

**Figure 1.** Comparison of the previous model and the model supported by our current data. In both the previous model (**A**) and the current model (**B**), long-chain ascarosides are shortened to medium- and short-chain ascarosides through β-oxidation cycles in the peroxisome. (**A**) According to the previous model, the final step in the biosynthesis of all of the IC-modified ascarosides is the attachment of the IC head group to the 4'-position of the ascarylose sugar. Short/medium-chain ascarosides, such as asc-C5 and asc-C9, are transported from the peroxisome to the lysosome, where the IC group is attached. According to this model, the acyl-CoA synthetase ACS-7 plays a key role in activating ICA as its corresponding CoA thioester, IC-CoA, for attachment specifically to asc-C5 to make IC-asc-C5. (**B**) According to our model, the IC head group is only attached to ascarosides with medium-length side chains, such as asc-C9. The biosynthesis of IC-modified ascarosides with shorter side chains, such as IC-asc-C5, requires additional β-oxidation in the peroxisome. In our model, under favorable conditions, the IC head group is attached by unknown enzymes to the medium-chain ascarosides, such as asc-C9, to make aggregation pheromones. This process may occur in the lysosome as it has been shown that the lysosome is required for the biosynthesis of 4'-modified ascarosides (*Panda et al., 2017*). If conditions decline, such as during starvation, ACS-7 then plays a key role for activating IC-asc-C9 as the corresponding CoA-thioester for further β-oxidation of its side chain in the peroxisome to make shorter-chain IC-ascarosides, such as the dauer pheromone IC-asc-C5.

DOI: https://doi.org/10.7554/eLife.33286.003

(icas#1) (*Panda et al., 2017*). Although we were also able to detect a peak on the LC-MS representing this conversion, the amount of IC-asc-C7 produced was so small relative to background that it could not be quantified. Thus, we conclude that *C. elegans* does not preferentially attach the IC group to ascarosides with C5 or C7 side chains. On the other hand, *C. elegans* can make the corresponding IC-ascarosides, IC-asc-C5 and IC-asc-C7. Therefore, we speculated that these short chain IC-ascarosides could potentially be made from medium-chain IC-ascarosides, such as IC-asc-C9, which have their side chains shortened through β-oxidation.

## Role of ACS-7 in activating medium-chain IC-ascarosides

The acyl-CoA synthetase ACS-7 has been reported to be required for biosynthesis of the short-chain IC-ascaroside, IC-asc-C5. Panda et al. hypothesized that this enzyme was involved in direct attachment of the IC group to the 4'-position of the ascarylose sugar of asc-C5 (*Figure 1A*) (*Panda et al., 2017*). They were able to show that ACS-7 activates ICA as the AMP-ester, IC-AMP (*Figure 1A*). However, they were not able to show that ACS-7 promotes conversion of IC-AMP to the corresponding CoA-thioester, IC-CoA, or that it leads to the reaction of IC-CoA with asc-C5 (*Figure 1A*).

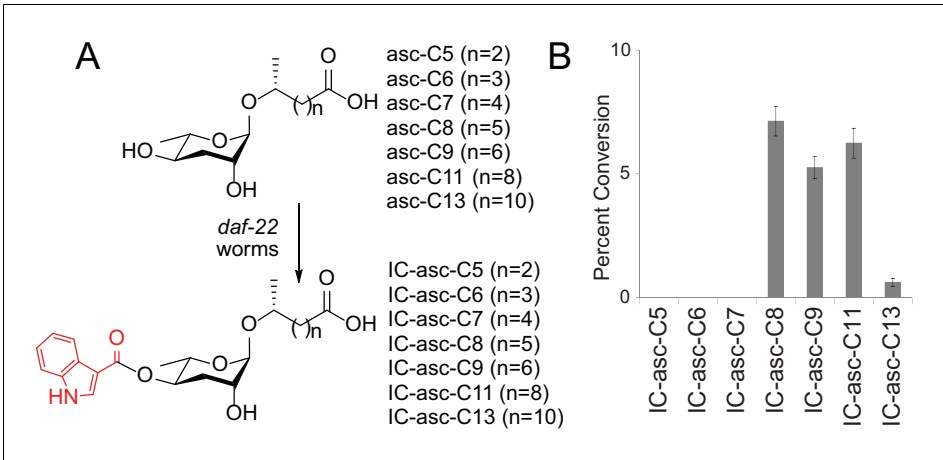

**Figure 2.** Substrate preference for attachment of the IC head group in biosynthesis of IC-ascarosides. (**A**) Possible attachment of the IC group to ascarosides with different side-chain lengths (C5–C13). (**B**) Percent conversion of each ascaroside to the corresponding IC-ascaroside by *daf-22(m130)* worms. Data represent the mean ± SD of three independent experiments.

DOI: https://doi.org/10.7554/eLife.33286.004

Furthermore, the $K_M$ of ACS-7 for ICA (in the one-step reaction to produce IC-AMP) is 270 ± 90 μM (**Panda et al., 2017**), which is much higher than the $K_M$ values of acyl-CoA synthetases from other organisms (in the two-step reaction to produce an acyl-CoA) that are generally in the low μM range (**Hall et al., 2003**; **Van Horn et al., 2005**).

Given our data showing that the IC group is not directly attached to asc-C5 or asc-C7 inside the worm, we considered alternative substrates for ACS-7. The *acs-7* mutant accumulates IC-asc-C9, while failing to make IC-asc-C5 (**Panda et al., 2017**). Therefore, we speculated that ACS-7 may activate, for example, IC-asc-C9, as its CoA-thioester, for subsequent β-oxidation of the side chain to make shorter-chain IC-ascarosides. To test this hypothesis, we purified recombinant His-tagged ACS-7 from *E. coli* and tested its activity against IC-asc-C9. ACS-7 showed robust activity towards this substrate, rapidly converting it completely to the corresponding CoA-thioester within 10 min (**Figure 3A**). We considered the possibility that recombinant ACS-7 may copurify with an enzyme from *E. coli* that might be responsible for this activity. Therefore, we designed as a negative control two ACS-7 catalytic mutants for testing against the IC-asc-C9 substrate. ACS-7 mutants in which the magnesium ion binding site was disrupted were designed through sequence alignment of ACS-7 with long-chain fatty acyl-CoA synthetase from *Thermus thermophilus*. The ACS-7(E339A) single mutant showed very little activity towards the IC-asc-C9 substrate, and the ACS-7(E339A, S186A, S187A) triple mutant showed no activity towards the IC-asc-C9 substrate, thereby confirming that the activity seen for the wild-type enzyme does not result from any copurifying proteins (**Figure 3A**).

ACS-7 showed very little activity towards ICA, the putative substrate reported by Panda et al. During the reaction of ACS-7 with ICA, we saw accumulation of the AMP-ester intermediate, suggesting that ACS-7 is very sluggish towards the ICA substrate and does not turn it over to the corresponding CoA-thioester (**Figure 3B**). Although we were able to detect the conversion of ICA to the corresponding CoA-thioester, the reaction required a 2-hour incubation (**Figure 3C**). Furthermore, based on the UV absorption at 280 nm, more than 60% of the substrate remained as the AMP-ester at this late time point.

To determine the substrate specificity of ACS-7, we tested it against a panel of substrates, including IC-ascarosides (IC-asc-C5, IC-asc-C7, IC-asc-ΔC9, and IC-asc-C9), ascarosides (asc-C7, asc-ΔC9, and asc-C9), and fatty acids (fatty acid-C7 and fatty acid-C9). These data show that ACS-7 strongly prefers IC-modified ascarosides over simple ascarosides, demonstrating that it is specifically involved in activation of IC-modified ascarosides for β-oxidation (**Figure 3D**). ACS-7 activates the longer-chain substrates (IC-asc-C7 and IC-asc-C9), but not short-chain substrates (IC-asc-C5), consistent with its role in shortening the longer-chain substrates. ACS-7 can activate fatty acids with medium-length side chains, but not short side chains. Although it is unclear whether the reaction of ACS-7 with fatty

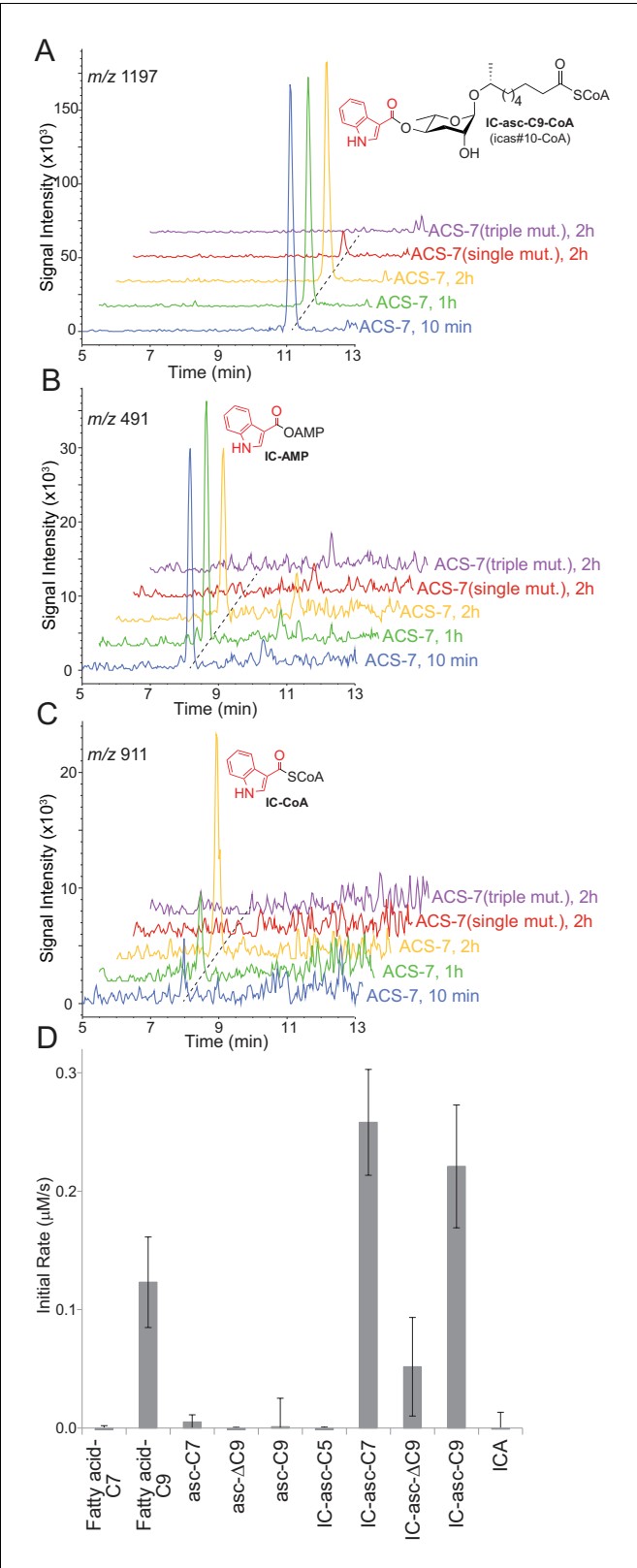

**Figure 3.** In vitro activity of ACS-7 towards ascaroside and ICA substrates. (**A**) LC-MS traces of the reaction of ACS-7, ACS-7(E339A) single mutant, or ACS-7(E339A, S186A, S187A) triple mutant with IC-asc-C9, detecting IC-asc-C9-CoA. (**B–C**) LC-MS traces of the reaction of ACS-7, ACS-7 (E339A), or ACS-7 (E339A, S186A, S187A) with ICA, detecting IC-AMP (**B**) or IC-CoA (**C**). (**D**) Activity of ACS-7 in a coupled enzyme assay against a panel of

*Figure 3 continued on next page*

*Figure 3 continued*

substrates, including fatty acids (fatty acid-C7 and fatty acid-C9), ascarosides (asc-C7, asc-ΔC9, and asc-C9), IC-modified ascarosides (IC-asc-C5, IC-asc-C7, IC-asc-ΔC9, and IC-asc-C9), and ICA. Data represent the mean ± SD of three independent experiments.

DOI: https://doi.org/10.7554/eLife.33286.005

acids is physiologically relevant, it could potentially indicate that ACS-7 also plays a role in fatty-acid metabolism, in addition to ascaroside biosynthesis. The $K_M$ of ACS-7 for IC-asc-C9 is $14.5 \pm 3.0$ μM, the $k_{cat}$ is $0.53 \pm 0.08$ s$^{-1}$, and the $k_{cat}/K_M$ is $36{,}300 \pm 9{,}200$ M$^{-1}$ s$^{-1}$.

## Role of *acox-1.1* and *acox-3* in processing of IC-ascarosides

Analysis of ascaroside production in acyl-CoA oxidase mutants suggests that β-oxidation does play a role in production of the IC-ascarosides. We have shown that ACOX-1.1 and ACOX-3 work together in the β-oxidation cycles that shorten ascarosides with 15-carbon and 13-carbon side chains (*Zhang et al., 2018*). Analysis of the ascarosides produced by *acox-1.1* and *acox-3* mutant worms indicates that these acyl-CoA oxidases also play a role in biosynthesis of short-chain IC-ascarosides. The amounts of the ascarosides and corresponding IC-ascarosides in wild-type, *acox-1.1*, *acox-3*, and *acox-1.1;acox-3* worms suggest that production of these two groups of ascarosides does not always correlate (*Figure 4*). Thus, it is unlikely that the IC group is directly added to short-chain ascarosides to make the corresponding IC-ascarosides. The *acox-1.1(ok2257)* deletion mutant shows an accumulation of IC-asc-C7, indicating that ACOX-1.1 may function in the β-oxidation cycle that

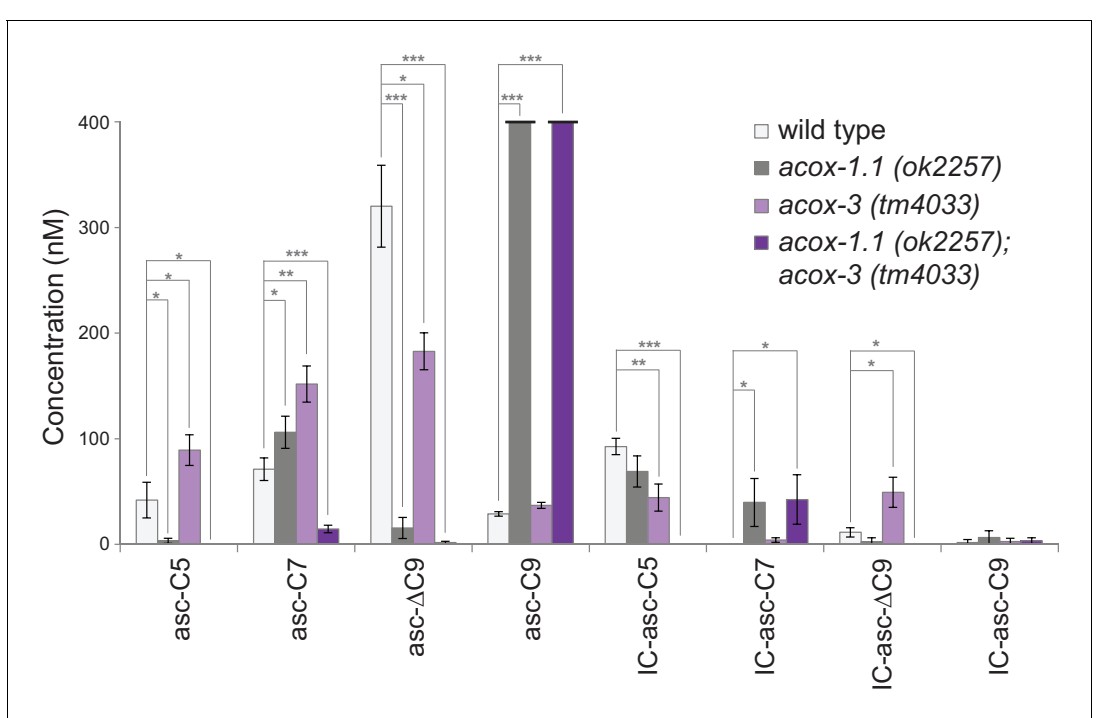

**Figure 4.** Ascaroside production in deletion mutants of acyl-CoA oxidase genes *acox-1.1* and *acox-3*. Comparison of ascaroside and corresponding IC-ascaroside production in wild type, the *acox-1.1(ok2257)* deletion mutant, the *acox-3(tm4033)* deletion mutant, and the *acox-1.1(ok2257);acox-3(tm4033)* double deletion mutant. Data represent the mean ± SD of three independent experiments. Two-tailed, unpaired t-tests were used to determine statistical significance (*p≤0.05, **p≤0.01, ***p≤0.001).

DOI: https://doi.org/10.7554/eLife.33286.006

The following figure supplement is available for figure 4:

**Figure supplement 1.** Ascaroside production in wild type, the *acox-1.1(reb2[E434A])* catalytic mutant, the *acox-1.4(reb6[E433A])* catalytic mutant, the *acox-3(tm4033)* deletion mutant, and double mutant and triple mutants.

DOI: https://doi.org/10.7554/eLife.33286.007

shortens a seven-carbon IC-ascaroside to a five-carbon IC-ascaroside (*Figure 4*). The *acox-3(tm4033)* deletion mutant shows decreased production of IC-asc-C5, but increased accumulation of IC-asc-C7 and IC-asc-ΔC9 (*Figure 4*, *Figure 4—figure supplement 1*). Thus, like ACOX-1.1, ACOX-3 may function in the β-oxidation cycle that shortens a seven-carbon IC-ascaroside to a five-carbon IC-ascaroside. The ascaroside profile of the *acox-1.1(ok2257);acox-3(tm4033)* double deletion mutant is quite striking in that production of IC-asc-C5 is completely abolished, while the amount of IC-asc-C7 is increased relative to wild type (*Figure 4*). The amount of IC-asc-C7 is increased despite the amount of asc-C7 being decreased, indicating that production of the ascarosides and IC-ascarosides are independently regulated. These data thus provide further evidence that ACOX-1.1 and ACOX-3 participate in a β-oxidation cycle that shortens IC-asc-C7 to IC-asc-C5.

We used CRISPR-Cas9 to generate an *acox-1.1(reb2[E434A])* catalytic mutant strain, in which the ACOX-1.1 enzyme is mutated in a glutamate in the active site that is important for catalytic activity (*Zhang et al., 2018*). This strain shows an accumulation of IC-asc-C7, indicating that ACOX-1.1's catalytic activity contributes to the shortening of IC-ascarosides (*Figure 4—figure supplement 1*). In addition to ACOX-1.1 and ACOX-3, ACOX-1.4 may also contribute to the β-oxidation of IC-ascarosides. An *acox-1.4(reb6[E433A])* catalytic mutant strain that we generated through CRISPR-Cas9, as well as an *acox-1.4(tm6415)* deletion mutant strain and an *acox-1.4(gk892586)* nonsense mutant strain, did not show defects in IC-ascaroside biosynthesis. However, examination of the *acox-1.1 (reb2[E434A]);acox-1.4(reb6[E433A]);acox-3(tm4033)* triple mutant shows that all three genes contribute to the β-oxidation cycle that shortens IC-asc-C9 to IC-asc-C7, as well as the β-oxidation cycle that shortens IC-asc-C7 to IC-asc-C5 (*Figure 4—figure supplement 1*).

## In vitro activity of ACOX-1.1 and ACOX-3 in β-oxidation of IC-ascarosides

To provide direct evidence for the role of ACOX-1.1 and ACOX-3 in biosynthesis of the IC-ascarosides, the two enzymes were expressed in *E. coli* and purified to assay their activity. Unfortunately, ACOX-1.4 could not be expressed in *E. coli* despite repeated attempts. In an LC-MS-based activity assay, the acyl-CoA oxidases, as well as MAOC-1, DHS-28, and DAF-22, were incubated with IC-asc-C9-CoA (*Figure 5A,B*; *Figure 5—figure supplement 1*). ACOX-1.1 and ACOX-3 both enabled the conversion of IC-asc-C9-CoA to IC-asc-C5-CoA. On the other hand, ACOX-1.2, which has a small active site and strictly prefers short-chain substrates (*Zhang et al., 2015*; *Zhang et al., 2016*), was inactive in this assay (*Figure 5A*). Under the assay conditions, the CoA-thioester bond was gradually cleaved, and thus, in the control and ACOX-1.2 reactions, unreacted IC-asc-C9-CoA substrate was gradually hydrolyzed to IC-asc-C9. Conversely, in the ACOX-1.1 and ACOX-3 reactions, IC-asc-C9-CoA was converted to IC-asc-C5-CoA, which was partially hydrolyzed to IC-asc-C5 (*Figure 5A*).

To further investigate the substrate preferences of the acyl-CoA oxidases, the enzymes were assayed in an enzyme-coupled assay in the presence of peroxidase, which can use the $H_2O_2$ produced by the reaction to generate a UV-active product. ACOX-1.1 was almost as active towards IC-asc-C7-CoA as it was towards its preferred ascaroside substrate, asc-C9-CoA (*Figure 5C*; *Figure 5—figure supplement 1*). It also showed activity towards IC-asc-C9-CoA, but less than towards IC-asc-C7-CoA. Unfortunately, ACOX-3 showed low activity towards all substrates tested in this assay, including another one of its preferred ascaroside substrates, asc-C13-CoA (*Zhang et al., 2018*). This result may suggest that ACOX-3 requires other β-oxidation enzymes to be fully active (*Figure 5—figure supplement 2*).

## Expression pattern of ACS-7, ACOX-1.1, and ACOX-3 in *C. elegans*

It has been shown previously that ACS-7 is expressed in the lysosome, where it was thought to contribute to IC-ascaroside biosynthesis (*Panda et al., 2017*). Because our data show that ACS-7 activates the side chains of medium-chain IC-ascarosides for peroxisomal β-oxidation, we hypothesized that ACS-7 must be expressed in the peroxisome. Indeed, ACS-7 has a PTS1-type peroxisomal localization signal. Furthermore, when we tested the enzymatic activity of ACS-7, we found that the enzyme was active at neutral pH, but not at the pH found in the lysosome (pH 5.0). To determine the localization pattern of ACS-7, we generated translational reporter constructs for ACS-7, ACOX-3, as well as ACOX-1.1, which was previously shown to be localized to the peroxisome (*Joo et al., 2010*). Co-injection of these constructs into wild-type *C. elegans* showed that all three proteins are

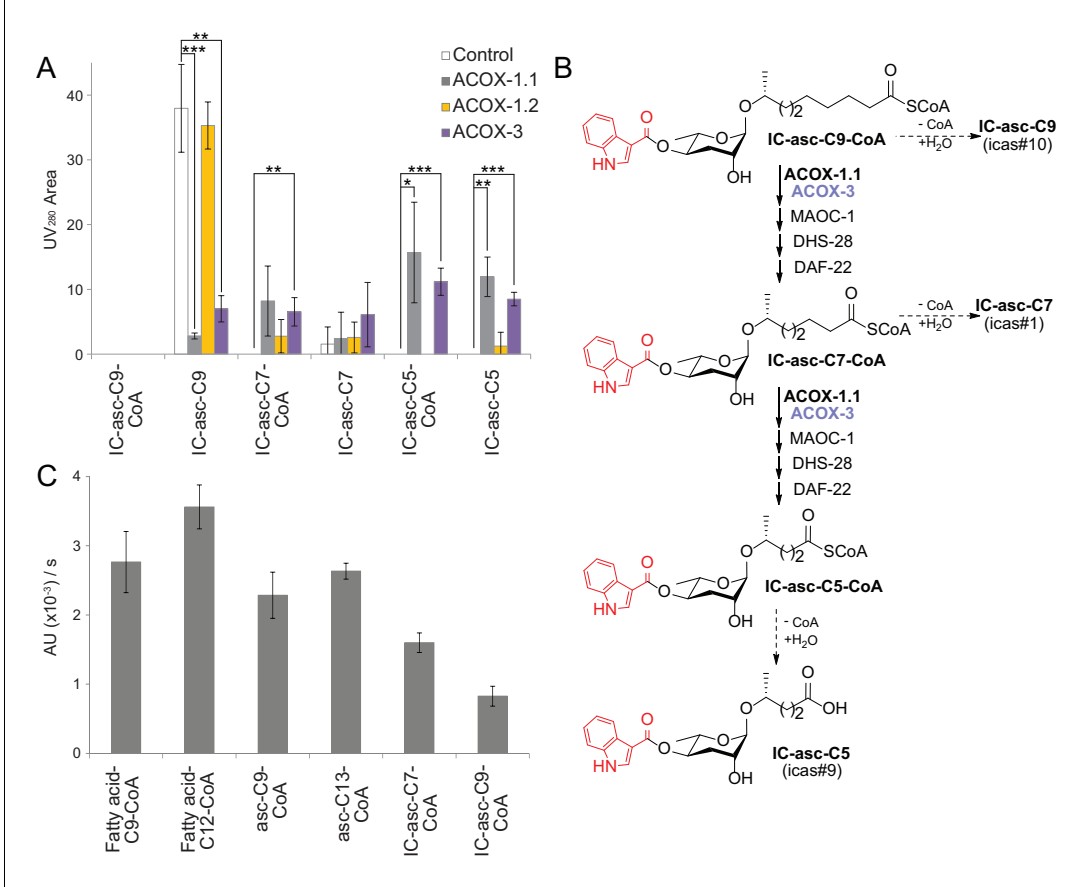

**Figure 5.** In vitro activity of ACOX-1.1, ACOX-1.2, and ACOX-3. (**A**) Reaction of IC-asc-C9-CoA with ACOX-1.1, ACOX-1.2, or ACOX-3, in the presence of the additional β-oxidation enzymes, MAOC-1, DHS-28, and DAF-22, as monitored by LC-MS. Chemical structures are shown in (**B**). Data represent the mean ± SD of three independent experiments. Two-tailed, unpaired t-tests were used to determine statistical significance (*p≤0.05, **p≤0.01, ***p≤0.001). (**B**) Proposed role for ACOX-1.1 and ACOX-3 in the β-oxidation of IC-asc-C9-CoA to IC-asc-C5-CoA. (**C**) Activity of ACOX-1.1 in a coupled enzyme assay against the CoA-thioesters of fatty acids (fatty acid-C9-CoA and fatty acid-C12-CoA), ascarosides (asc-C9-CoA and asc-C13-CoA), and IC-modified ascarosides (IC-asc-C7-CoA and IC-asc-C9-CoA). Data represent the mean ± SD of three independent experiments.
DOI: https://doi.org/10.7554/eLife.33286.008

The following figure supplements are available for figure 5:

**Figure supplement 1.** High-resolution LC-MS/MS analysis of CoA-thioesters of ascarosides produced chemoenzymatically and used as substrates in *Figure 5A,C*.
DOI: https://doi.org/10.7554/eLife.33286.009

**Figure supplement 2.** Activity of ACOX-3 towards different substrates.
DOI: https://doi.org/10.7554/eLife.33286.010

expressed in a punctate pattern in the intestine (*Figure 6A,B*). In addition, the intestinal expression pattern of ACS-7 overlaps with that of ACOX-1.1 and ACOX-3 (*Figure 6A,B*). Some GFP signal is occasionally seen in LROs, especially in older worms (*Figure 6—figure supplement 1A,B*). However, we believe that this GFP signal is caused by autofluorescence of the LROs (*Hermann et al., 2005*) because (1) it only becomes prominent on prolonged bleaching of the worms under the fluorescent microscope, and (2) it is also observed in wild-type (N2) worms that have not been injected with the *Pasc-7::gfp::acs-7* reporter. (*Figure 6—figure supplement 1C–E*). RNAi against *prx-13*, which is a component of the peroxisomal import machinery (*Thieringer et al., 2003*), leads to a diffuse pattern of expression for ACOX-1.1 and ACS-7 in the intestine, providing further evidence that these enzymes are localized to the peroxisome (*Figure 6C,D*). Thus, ACS-7 likely participates in biosynthesis of the IC-ascarosides in the peroxisome rather than lysosome.

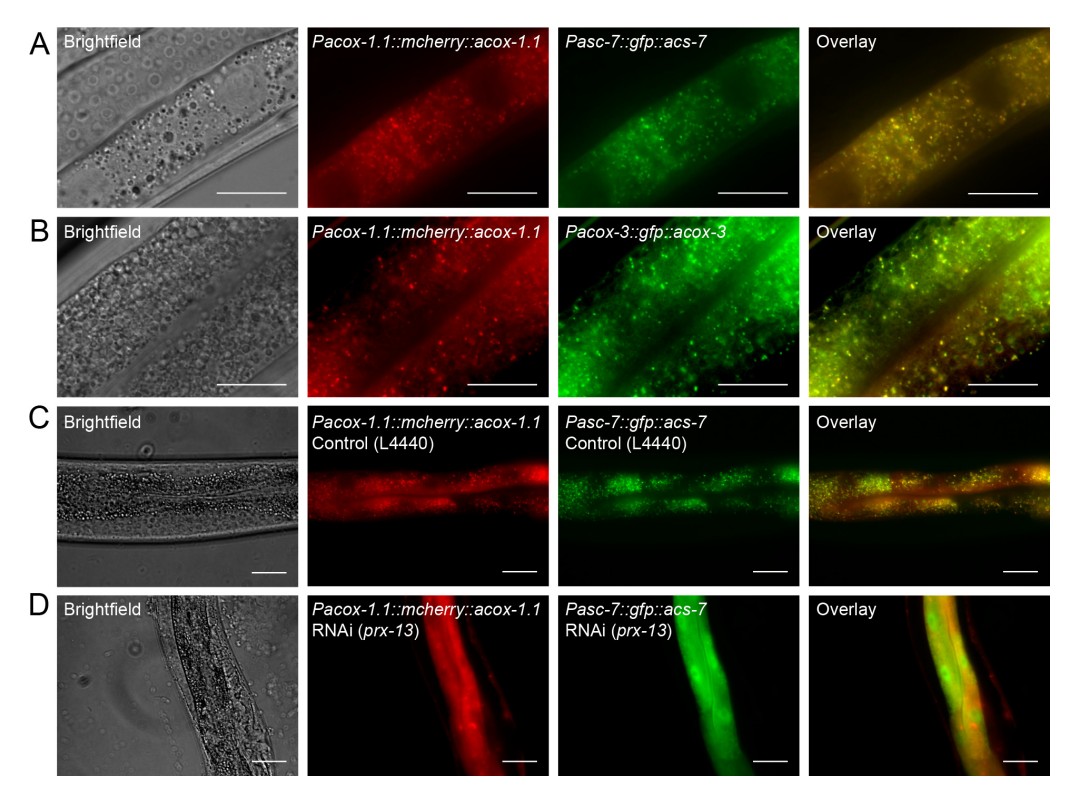

**Figure 6.** Peroxisomal co-localization of ACS-7, ACOX-1.1, and ACOX-3. (**A**) Co-injection of plasmids containing *Pacox-1.1::mcherry::acox-1.1* and *Pacs-7::gfp::acs-7* into wild-type worms shows that ACOX-1.1 and ACS-7 co-localize in a punctate pattern in the intestine. (**B**) Co-injection of plasmids containing *Pacox-1.1::mcherry::acox-1.1* and *Pacox-3::gfp::acox-3* into wild-type worms shows that ACOX-1.1 and ACOX-3 co-localize in a punctate pattern in the intestine. (**C**) RNAi using the control plasmid L4440 in *Pacox-1.1::mcherry::acox-1.1;Pacs-7::gfp::acs-7* worms gives a punctate pattern of expression in the intestine. (**D**) RNAi against *prx-13* in *Pacox-1.1::mcherry::acox-1.1; Pacs-7::gfp::acs-7* worms gives a diffuse pattern of expression in the intestine. In (**A–D**), L4 to young adult stage worms were imaged. Scale bar = 20 µm.

DOI: https://doi.org/10.7554/eLife.33286.011

The following figure supplement is available for figure 6:

**Figure supplement 1.** ACS-7 does not localize primarily to the lysosome.

DOI: https://doi.org/10.7554/eLife.33286.012

## Discussion

Our data show that *C. elegans* can engage in dynamic tailoring of the ascarosides that it has produced. This dynamic tailoring enables the worm to respond rapidly to changing environmental conditions and modulate the nature of its chemical message without having to synthesize new ascarosides de novo. Specifically, we have shown that *C. elegans* attaches the IC head group to ascarosides with medium-length (8–11) side chains to generate aggregation pheromones such as IC-asc-ΔC9 (icas#3) (*Srinivasan et al., 2012*). Production of aggregation pheromones likely occurs under favorable environmental conditions such as food-rich conditions as *C. elegans* tends to aggregate on food (*Srinivasan et al., 2012*). We have shown that *C. elegans* can activate IC-ascarosides with medium-length side chains for shortening of the side chains through β-oxidation to generate the dauer pheromone IC-asc-C5 (icas#9) (*Figure 1B*). This β-oxidation process likely occurs when environmental conditions become less favorable, such as during starvation. Indeed, starvation conditions have been shown to lead to an increase in the short-chain IC-ascaroside IC-asc-C5 in L1 larvae (*Artyukhin et al., 2013*). Consistent with this result, bacterial food has been shown to downregulate production of IC-asc-C5 in L4 larvae (*Zhang et al., 2015*). Increased production of short-chain IC-ascarosides by starvation conditions would be consistent with their biological role as IC-asc-C5 is a

dauer pheromone (*Butcher et al., 2009a*), and *C. elegans* would likely want to enter dauer under starvation conditions. Our work has thus determined the mechanism by which *C. elegans* modulates the balance between aggregation-inducing, medium-chain IC-ascarosides and dauer-inducing, short-chain IC-ascarosides in response to changing environmental conditions.

Our model for biosynthesis of the short-chain IC-ascarosides, in which β-oxidation of the ascaroside side chain occurs after attachment of the IC head group, contrasts with the model proposed by Panda et al., in which β-oxidation of the ascaroside side chain occurs before attachment of the IC head group (*Panda et al., 2017*) (*Figure 1A,B*). We have shown that the acyl-CoA synthetase ACS-7 activates the side chains of medium-chain IC-ascarosides, such as IC-asc-C9, as CoA-thioesters, thereby enabling the side chains to be shortened through β-oxidation (*Figure 1B*). Previously, Panda et al. proposed that ACS-7 activates ICA as the corresponding CoA-thioester, IC-CoA, and enables attachment of the IC group to asc-C5 to make IC-asc-C5 (*Panda et al., 2017*) (*Figure 1A*). However, ACS-7 could only be shown to convert ICA to IC-AMP (*Panda et al., 2017*). Our data show that IC-asc-C9 is a much better substrate for ACS-7, which rapidly converts IC-asc-C9 to the corresponding CoA-thioester, IC-asc-C9-CoA (*Figure 1B*). We have shown that once the side chain of the medium-chain IC-ascarosides is activated, it can be shortened through β-oxidation cycles involving ACOX-1.1, ACOX-3, and likely ACOX-1.4 as well, to make the short-chain IC-ascaroside, IC-asc-C5. Specifically, we have shown that the *acox-1.1;acox-3* double deletion mutant fails to make any IC-asc-C5 and accumulates IC-asc-C7. Furthermore, ACOX-1.1 and ACOX-3 are able to process IC-asc-C9-CoA to IC-asc-C5-CoA in vitro with the help of the β-oxidation enzymes MAOC-1, DHS-28, and DAF-22. Consistent with ACS-7 working in the same biosynthetic pathway as ACOX-1.1 and ACOX-3, the three genes appear to be transcriptionally co-regulated (*von Mering et al., 2005*). ACOX-3 has been shown to be induced by dietary restriction (*Palgunow et al., 2012*). ACS-7 and ACOX-3 have also been shown to be co-regulated by the transcription factor SKN-1, which functions in a variety of stress responses, including detoxification, pathogen defense, the unfolded protein response, and lipid metabolism under starvation conditions (*Blackwell et al., 2015*). However, the significance of this regulation for IC-ascaroside biosynthesis remains to be seen.

In addition to the IC-ascarosides, ACS-7 is also necessary for biosynthesis of short-chain OS-ascarosides, as *acs-7* mutant worms cannot make the short-chain OS-ascarosides (*Panda et al., 2017*). Thus, it is likely that, in addition to medium-chain IC-ascarosides, ACS-7 also activates medium-chain OS-ascarosides for β-oxidation of their side chains. The β-oxidation of IC- and OS-modified ascarosides may be regulated in a similar fashion. Starvation has been shown to increase the ratio of short-chain (C5) to medium-chain (C9) OS-ascarosides (*Artyukhin et al., 2013*). Production of the short-chain OS-ascaroside OS-asc-C5 (osas#9) under starvation conditions is consistent with its biological role, as it acts as a dispersal signal that encourages the worm population to seek more favorable conditions (*Artyukhin et al., 2013*). In addition to the IC and OS head groups, the 4'-position of the ascarosides can also be modified by the MB and HB head groups (*von Reuss et al., 2012*). However, MB- and HB-ascarosides with fewer than nine-carbons in their side chains have not been detected in *C. elegans* (*von Reuss et al., 2012*). Thus, unlike medium-chain IC-and OS-ascarosides, medium-chain MB- and HB-ascarosides probably do not undergo activation by an acyl-CoA synthetase for further β-oxidation in the peroxisome.

In contrast to previous reports that suggested ACS-7 is localized to the lysosomes in the intestine (*Panda et al., 2017*), we have shown that ACS-7 is localized to the peroxisomes. It is likely that the peroxisome is where ACS-7 contributes to IC-ascaroside biosynthesis, as ACS-7 activates the medium-chain IC-ascarosides as their CoA-thioesters for β-oxidation by the peroxisomal enzymes ACOX-1.1/ACOX-3, MAOC-1, DHS-28, and DAF-22. The lysosome has been shown to be important for biosynthesis of ascarosides modified with a head group on the 4'-position of the ascarylose sugar (*Panda et al., 2017*). Mutant worms that lack LROs, such as *glo-1* mutants, do not make any ascarosides modified on the 4'-position with IC, OS, HB, or MB head groups (*Panda et al., 2017*). In a model that accounts for the role of the LROs in ascaroside biosynthesis, long-chain ascarosides could be shortened to medium- and short-chain ascarosides in the peroxisome, and then medium-chain ascarosides, such as asc-C9, may be transported from the peroxisome to the LROs for attachment of 4'-modifications, such as attachment of the IC head group to make IC-asc-C9, by unknown enzymes (*Figure 1B*). In response to changing internal or external conditions, these modified ascarosides could then be transported back to the peroxisome for activation by ACS-7 and further β-oxidation of their side chains (*Figure 1B*). Trafficking of the ascarosides through multiple cellular organelles,

such as the peroxisome and LROs, during biosynthesis may enable *C. elegans* to induce the production of modified ascarosides with specific side-chain lengths only under certain conditions or only in certain tissues.

# Materials and methods

## Key resources table

| Reagent type (species) or resource | Designation | Source or reference | Identifiers | Additional information |
|---|---|---|---|---|
| Gene (*C. elegans*) | *acox-1.1* | | WBGene00008564 | Other names: CELE_F08A8.1, *acox-1* |
| | *acox-1.2* | | WBGene00008565 | Other names: CELE_F08A8.2, *acox-2, drd-100* |
| | *acox-1.3* | | WBGene00008566 | Other names: CELE_F08A8.3, *acox-3* |
| | *acox-1.4* | | WBGene00008567 | Other names: CELE_F08A8.4, *acox-4* |
| | *acox-1.5* | | WBGene00008167 | Other names: CELE_C48B4.1, *acox-5, drd-51* |
| | *acox-1.6* | | WBGene00010336 | Other names: CELE_F59F4.1, *acox-6* |
| | *acox-3* | | WBGene00019060 | Other names: CELE_F58F9.7, *acox-6* |
| | *acs-7* | | WBGene00007228 | Other name: CELE_C01G6.7 |
| | *maoc-1* | | WBGene00017123 | Other name: CELE_E04F6.3 |
| | *dhs-28* | | WBGene00000991 | Other name: CELE_M03A8.1 |
| | *daf-22* | | WBGene00013284 | Other name: CELE_Y57A10C.6 |
| Gene (*M. tuberculosis*) | *fadD6* | | Gene ID: 887549 | |

*Continued on next page*

*Continued*

| Reagent type (species) or resource | Designation | Source or reference | Identifiers | Additional information |
|---|---|---|---|---|
| Strains (*C. elegans*) | N2 | CGC | RRID:WB-STRAIN:N2_(ancestral) | wild type (Bristol) |
| | RAB1 | doi: 10.1021/acschembio.7b01021 | | backcrossed VC1785 *acox-1.1 (ok2257)*I |
| | RAB21 | doi: 10.1021/acschembio.7b01021 | | backcrossed *acox-1.4(tm6415)*I |
| | VC40449 | CGC | RRID:WB-STRAIN:VC40449 | *acox-1.4(gk892586)*I |
| | RAB22 | doi: 10.1021/acschembio.7b01021 | | backcrossed *acox-3(tm4033)*IV |
| | DR476 | CGC | RRID:WB-STRAIN:DR476 | *daf-22(m130)*II |
| | RAB24 | doi: 10.1021/acschembio.7b01021 | | *acox-1.1(reb2[E434A])*I |
| | RAB28 | doi: 10.1021/acschembio.7b01021 | | *acox-1.4(reb6[E433A])*I |
| | RAB30 | doi: 10.1021/acschembio.7b01021 | | *acox-1.1(ok2257)*I; *acox-3(tm4033)*IV |
| | RAB31 | doi: 10.1021/acschembio.7b01021 | | *acox-1.1(reb2[E434A])*I; *acox-1.4 (reb6[E433A])*I |
| | RAB32 | doi: 10.1021/acschembio.7b01021 | | *acox-1.1(reb2[E434A])*I; *acox-3 (tm4033)*IV |
| | RAB35 | this paper | | *acox-1.4(reb6[E433A])*I; *acox-3 (tm4033)*IV |
| | RAB36 | this paper | | *acox-1.1(reb2[E434A])*I; *acox-1.4 (reb6[E433A])*I; *acox-3(tm4033)*IV |
| | RAB37 | this paper | | *rebEx11 (Pacs-7::gfp::acs-7)* |
| | RAB38 | this paper | | *rebEx12 (Pacox-1.1::mcherry:: acox-1.1; Pacs-7::gfp::acs-7)* |
| | RAB39 | this paper | | *rebEx13 (Pacox-1.1::mcherry:: acox-1.1; Pacox-3::gfp::acox-3)* |
| | RAB40 | this paper | | *rebEx14 (Pacox-1.1::mcherry:: acox-1.1; Pacs-7::gfp::acs-7)* |
| Plasmids for protein expression | pET-22b-*acs-7* | this paper | | Cloning described in 'ACS-7 expression and mutagenesis' |
| | pET-22b-*acs-7(E339A)* | this paper | | Cloning described in 'ACS-7 expression and mutagenesis' |
| | pET-22b-*acs-7(E339A, S186A,S187A)* | this paper | | Cloning described in 'ACS-7 expression and mutagenesis' |
| | pET-16b-*acox-1.1a* | doi: 10.1073/pnas.1423951112 | | |
| | pET-16b-*acox-1.2* | doi: 10.1073/pnas.1423951112 | | |
| | pET-16b-*acox-3* | this paper | | Cloning described in 'β-oxidation enzyme expression' |
| | pACYCDuet-1-*maoc-1* | this paper | | Cloning described in 'β-oxidation enzyme expression' |
| | pACYCDuet-1-*dhs-28Δscp-2* | this paper | | Cloning described in 'β-oxidation enzyme expression' |
| | pET-16b-*daf-22* | this paper | | Cloning described in 'β-oxidation enzyme expression' |
| | pET-28a-*fadD6* | doi: 10.1021/acschembio.7b01021 | | Cloning described in 'Synthesis of CoA-thioesters' |

*Continued*

| Reagent type (species) or resource | Designation | Source or reference | Identifiers | Additional information |
|---|---|---|---|---|
| Plasmids for generating transgenic strains | pPD114.108-*Pacox-1.1::mcherry::acox-1.1* | this paper | | Cloning described in 'ACOX-1.1, ACOX-3, and ACS-7 localization and RNAi' |
| | pPD114.108-*Pacs-7::gfp::acs-7* | this paper | | Cloning described in 'ACOX-1.1, ACOX-3, and ACS-7 localization and RNAi' |
| | pPD114.108-*Pacox-3::gfp::acox-3* | this paper | | Cloning described in 'ACOX-1.1, ACOX-3, and ACS-7 localization and RNAi' |
| Commercial kit | Q5 Site-Directed Mutagenesis Kit | New England Biolabs | E0554S | |
| Commercial enzymes | adenylate kinase | Sigma | M3003 | |
| | pyruvate kinase and lactate dehydrogenase | Sigma | P0294 | |
| Commercial compounds | LysoTracker Red (Deep Red) | Thermo Fisher | L12492 | |

## *C. elegans* strains

The following strains were used: wild-type (N2, Bristol), RAB1 *acox-1.1(ok2257)* I, RAB24 *acox-1.1 (reb2[E434A])* I, RAB28 *acox-1.4(reb6[E433A])* I, RAB21 *acox-1.4(tm6415)* I, VC40944 *acox-1.4 (gk892586)* I, RAB22 *acox-3(tm4033)* IV, RAB30 *acox-1.1(ok2257);acox-3(tm4033)*, RAB31 *acox-1.1 (reb2[E434A]);acox-1.4(reb6[E433A])*, RAB32 *acox-1.1(reb2[E434A]);acox-3(tm4033)*, RAB35 *acox-1.4 (reb6[E433A]);acox-3(tm4033)*, RAB36 *acox-1.1(reb2[E434A]);acox-1.4(reb6[E433A]); acox-3(tm4033)*, DR476 *daf-22(m130)* II. The *acox-1.1(ok2257)*, *acox-1.4(tm6415)*, and *acox-3(tm4033)* strains were backcrossed four or six times. The *acox-1.1(reb2[E434A])*, *acox-1.4(reb6[E433A])*, and *acox-1.1(reb2 [E434A]);acox-1.4(reb6[E433A])* mutants were generated using CRISPR-Cas9 and backcrossed two to six times (*Zhang et al., 2018*). All CRISPR-Cas9 mutants were made using the Fire laboratory's marker-free CRISPR protocol (*Arribere et al., 2014*; *Kim et al., 2014*; *Cong and Zhang, 2015*; *Farboud and Meyer, 2015*).

## In vivo attachment of the IC group to ascarosides

Cultures were grown similar to a previously published method (*von Reuss et al., 2012*), but with modifications. *daf-22(m130)* worms from one 10 cm NGM plate (seeded with 0.75 mL 25X OP50) were collected after the bacteria were completely consumed and transferred to a 25 mL pre-culture, feeding with 10 mL 25X OP50 on day 1 and day 3. On day 4, 1.25 mL of the pre-culture was transferred to 3.75 mL fresh S medium to start the ascaroside-supplemented cultures. Each of the 5 mL cultures were supplemented with 10 µL of 3 mM asc-C5, asc-C6, asc-C7, asc-C8, asc-C9, asc-C11, or asc-C13. 10 µL of ethanol was used for control. The supplemented cultures were fed with 1 mL of 25X OP50 each day for 4 days, and the culture medium was collected on the fifth day. 1 mL of the culture medium was lyophilized and extracted with 100 µL 50% methanol/water, and 10 µL of the supernatant was analyzed by LC-MS (*Zhang et al., 2013*). Given that detection of IC-ascarosides in culture medium by LC-MS suffers from ion suppression, a calibration curve of pure, synthetic IC-asc-ΔC9 in culture medium extract was generated as follows: 2.8 µL 600 µM IC-asc-ΔC9 standard was mixed with 57.2 µL *daf-22* culture medium extract (in 50% MeOH/water) to make a 28 µM stock. A series of 1:2 diluted standards was generated by mixing the stock 1:1 with *daf-22* culture medium extract. 5 µL of the standards (each containing 4.4, 8.8, 17.5, 35, 70, or 140 pmol IC-asc-ΔC9, respectively) were injected into the LC-MS on the same day of sample analysis. Trend lines describing the relationship between peak area and IC-asc-ΔC9 amount were used to analyze the molar amount of all IC-ascarosides present in the supplemented *daf-22* cultures. Three independent experiments were performed using worms cultured at three different times.

## ACS-7 expression and mutagenesis

 acs-7 was cloned from a *C. elegans* cDNA library by PCR and ligated into pET-22b using the NdeI and XhoI restriction sites. To engineer an appropriate negative control for enzyme assays, catalytic mutants of ACS-7 were designed through sequence alignment with ttLC-FACS (*Hisanaga et al., 2004*) from *Thermus thermophiles*. pET-22b-*acs-7(E339A)* was obtained via Q5 Site-Directed Mutagenesis Kit (New England Biolabs, Ipswich, MA) using pET-22b-*acs-7* as the template, and it was then further modified by the kit to generate pET-22b-*acs-7(E339A,S186A,S187A)*. Wild-type ACS-7 and the mutants were expressed in BL21 (DE3) cells by inducing with 0.3 mM IPTG at 16°C for 42 hr. The enzymes were purified by lysing the cells in lysis buffer (50 mM $KPO_4$ pH 7.4, 100 mM KCl) using a microfluidizer, incubating the lysate with Ni-NTA resin (Thermo Fisher, Waltham, MA), and eluting the resin with lysis buffer containing 500 mM imidazole. The enzymes were further purified through FPLC on a HiLoad 16/600 Superdex 200 column (GE Healthcare, Chicago, IL).

## LC-MS-based ACS-7 activity assay

Purified ACS-7, ACS-7(E339A), and ACS-7(E339A,S186A,S187A) were concentrated to 2 mg/mL. 2 μL of the 2 mg/mL enzyme was used in each 50 μL reaction mixture, which gave a 1.57 μM final concentration. The reaction mixture contained 100 mM $KPO_4$ pH 7.0, 5 mM ATP, 5 mM $MgCl_2$, 5 mM CoA, and 100 μM IC-asc-C9 or ICA as substrates. Reaction mixtures were incubated at 25°C for 10 min, 1 hr, or 2 hr. 50 μL of methanol was added to the reaction mixtures to quench them, and 5 μL of the 100 μL 1:1 reaction/methanol mixture was analyzed by LC-MS directly. LC-MS analysis was performed with an Agilent 6130 single quadrupole mass spectrometer, operating in both positive and negative modes, using a method adapted from a previously published one (*Zhang et al., 2013*). The LC conditions were holding for 2 min at 95% solvent A (water with 10 mM ammonium acetate) and 5% solvent B (acetonitrile), followed by gradually ramping up to 100% solvent B over 24 min. The MS was operated in full-scan mode ($m/z$ 150–1500) with a fragmentor voltage of 125 V, peak width of 0.15 min, and cycle length of 2.20 s/cycle.

## ACS-7-coupled enzyme assay

The enzyme kinetics of ACS-7 were determined through an enzyme-coupled spectrophotometric assay that measures the release of AMP (*Tanaka et al., 1979*; *Hisanaga et al., 2004*). Each 100 μL assay mixture contained 0.1 M Tris-HCl pH 7.4, 5 mM dithiothreitol, 1.6 mM Triton X-100, 10 mM $MgCl_2$, 7.5 mM ATP, 0.2 mM phosphoenol pyruvate, 0.15 mM NADH, 2 μL of adenylate kinase solution (Sigma M3003, prepared according to the manufacturer's protocol), 2 μL of pyruvate kinase and lactate dehydrogenase mixture stock (Sigma P0294), 4 μg of ACS-7, and the tested substrates with concentrations from 5 to 250 μM. 1 mM final concentration of CoA was added to the assay mixture to start the reaction. The reaction was run at 22°C, and the UV absorbance at 340 nm was measured to determine the activity of ACS-7. Three independent experiments were performed using protein purified at three different times.

## Ascaroside analysis of large-scale cultures

Large-scale (150 mL) non-synchronized worm cultures were fed *E. coli* (HB101) and grown for 9 d, and extracts were generated from the culture medium, as described (*Zhang et al., 2013*). Three independent experiments were performed using worms cultured at three different times. LC-MS/MS analysis of ascarosides from extracts was performed as described (*Zhang et al., 2015*), but with some modifications. A Phenomenex Kinetex 2.6 μM $C_{18}$ 100 Å (100 × 2.1 mm) column was attached to an Accela UHPLC and a Thermo TSQ Quantum Max mass spectrometer, operating in negative ion, heated (H)-ESI, precursor scanning mode (selecting for a product ion of $m/z$ 73.0). Quantitation of ascarosides by LC-MS/MS was done by generating a calibration curve using synthetic standards. All ascarosides were quantified using their corresponding synthetic standard, except for IC-asc-C7 and IC-asc-C9, which were quantitated using synthetic IC-asc-ΔC9.

## Ascaroside analysis of small-scale cultures

For *Figure 4—figure supplement 1*, small-scale (5 mL) non-synchronized worm cultures were started with worms from one 6 cm NGM-agar plate, fed *E. coli* (HB101), and grown for 7 days. Three independent experiments were performed using worms cultured at three different times. 5 mL of culture

was centrifuged (800 g for 2 min), the worms at the bottom were removed, and the supernatant was centrifuged again (3500 rpm for 10 min). 1 mL of this supernatant was lyophilized and resuspended in 100 μL of 50% methanol in water, and the ascarosides were analyzed by LC-MS as described (*Zhang et al., 2013*). LC-MS analysis of ascarosides was performed on a Phenomenex Luna 5 μm $C_{18}$ 2 100 Å (100 × 4.6 mm) column attached to an Agilent 1260 infinity binary pump and Agilent 6130 single quad mass spectrometer with API-ES source, operating in dual negative/positive single-ion monitoring mode, as previously described (*Zhang et al., 2015*). In general, all ascarosides were detected by LC-MS using the [M-H]$^-$ ion.

## Synthesis of CoA-thioesters

Ascarosides and IC-ascarosides were synthesized as previously described (*Hollister et al., 2013*), except that for the IC-ascarosides, the final reaction products were purified by Agilent 1200 Series HPLC on a Supelco Discovery 10 μm $C_{18}$ (250 × 10 mm) column. A water (with 0.1% formic acid) and acetonitrile (with 0.1% formic acid) solvent gradient was used, starting from 5% acetonitrile, ramping to 100% acetonitrile over 25 min, and then holding at 100% acetonitrile for 4 min, with a flow rate of 2 mL / min.

CoA-thioesters of fatty acids and short- and medium-chain ascarosides were synthesized as previously described with several modifications (*Zhang et al., 2015*). Ascaroside (3 μM) was dissolved in tetrahydrofuran (350 μL), followed by the addition of carbonyldiimidazole (4.5 ~ 6 μM) in tetrahydrofuran (90 μL). The reaction was allowed to stir at room temperature for 1 hr and then dried under $N_2$ gas or by speedvac. The mixture was then dissolved in tetrahydrofuran (400 μL) followed by adding CoA (3 μM) in water (200 μL). The reaction was allowed to stir at room temperature for 4 hr. Dried product was purified by Agilent 1260 Infinity HPLC on a Supelco Discovery 10 μm $C_{18}$ (250 × 21.2 mm) column. A water (with 10 mM ammonium acetate) and acetonitrile solvent gradient was used, starting from 0% acetonitrile, ramping to 80% acetonitrile over 25 min, and then holding at 100% acetonitrile for 2 min, with a flow rate of 8 mL / min.

To synthesize the CoA-thioesters of long-chain ascarosides or IC-ascarosides, a chemoenzymatic approach was taken using the fatty acyl-CoA ligase FadD6 (*Arora et al., 2005*) or ACS-7. The *fadD6* gene was cloned from a *Mycobacterium tuberculosis* H37Ra genomic library (a gift from Peilan Zhang and Prof. Yousong Ding) and expressed as described (*Zhang et al., 2015*; *Zhang et al., 2018*). FadD6 reaction conditions were similar to previously described conditions (*Arora et al., 2005*) with some modifications. For a 200 μL total reaction volume, ~300 μM asc-C13, IC-asc-C7, or IC-asc-C9, 5 mM CoA, 15 mM ATP and 20 μg FadD6 protein were added to reaction buffer (100 mM Tris, 8 mM $MgCl_2$, pH 7.5), and the reaction was incubated at 30°C for 1 hr. The asc-C13-CoA, IC-asc-C7-CoA, and IC-asc-C9-CoA were purified by Agilent 1200 Series HPLC on a Phenomenex Luna 5 μm $C_{18}$ 2 100 Å (100 × 4.6 mm) column. A water (with 10 mM ammonium acetate) and acetonitrile solvent gradient was used, starting from 0% acetonitrile, ramping to 100% acetonitrile over 20 min, and then holding at 100% acetonitrile for 2 min, with a flow rate of 0.7 mL / min. Purified asc-C13-CoA, IC-asc-C7-CoA, and IC-asc-C9-CoA were then dried with a speedvac. Although FadD6 was initially used in the chemoenzymatic synthesis of these molecules, once we identified the substrate preferences of ACS-7, we subsequently used it to make IC-asc-C7-CoA and IC-asc-C9-CoA, as it was much faster and far more consistent than FadD6 at making these compounds.

## Characterization of CoA-thioesters

The structures of synthetic IC-asc-C7-CoA, IC-asc-C9-CoA, and asc-C13-CoA were confirmed by LC-MS/MS, using a Nano LCMS Solutions 3 μM 200 Å (0.3 × 150 mm) ProtoSIL C18AQ + column attached to an UltiMate 3000 RSLCnano System and a Bruker Impact II QTOF mass spectrometer, operating in positive ion, heated (H)-ESI mode. A water (with 100 mM ammonium acetate) and acetonitrile (with 0.1% formic acid) solvent gradient was used for separation, starting from 2% acetonitrile for 7 min, ramping to 98% acetonitrile over 31 min, and then holding at 98% acetonitrile for 10 min, with a flow rate of 5 μL / min. HR-MS/MS analysis was applied to all three samples with collision energy set at 45 eV. Characteristic fragmentation was observed, such as a neutral loss of 507 (*Magnes et al., 2005*) (*Figure 5—figure supplement 1*). IC-asc-C7-CoA, HR-ESIMS (*m/z*): [M + H]$^+$ calcd. for $C_{43}H_{64}N_8O_{22}P_3S$ 1169.3068, found 1169.3105. IC-asc-C9-CoA, HR-ESIMS (*m/z*): [M + H]$^+$

calcd. for $C_{45}H_{68}N_8O_{22}P_3S$ 1197.3382, found 1197.3390. asc-C13-CoA, HR-ESIMS ($m/z$): $[M + H]^+$ calcd. for $C_{40}H_{71}N_7O_{21}P_3S$ 1110.3637, found 1110.3712.

## β-oxidation enzyme expression

The cloning and expression conditions for the ACOX-1.1a (the longest splice variant of ACOX-1.1) homodimer were described previously (*Zhang et al., 2015*). *Acox-3* was cloned by PCR from a *C. elegans* (N2) cDNA library and was inserted into a modified pET-16b vector at the NheI/NotI restriction sites (resulting in a C-terminal His tag). The plasmid was transformed into BL21(DE3) cells, and a culture was grown at 37°C until the $OD_{600}$ reached 0.7, at which point expression was induced overnight at 25°C using 0.6 mM IPTG. The cells were resuspended in buffer (25 mM Tris, pH 7.5, 500 mM NaCl, 20 µM FAD) and lysed using a microfluidizer, and the protein was purified with Ni-NTA resin (Thermo Fisher). After the protein was concentrated with a 10 KDa cut-off centricon (MilliporeSigma, Burlington, MA), it was further purified through FPLC on a HiLoad 16/600 Superdex 200 column (GE Healthcare) and concentrated again to 1 mg/mL for assay. *maoc-1*, *dhs-28*, and *daf-22* genes were cloned by PCR from a *C. elegans* (N2) cDNA library. The *maoc-1* gene was inserted into the pACYCDuet-1 vector at the EcoRI/NotI sites (resulting in an N-terminal His tag) to generate pACYCDuet-1-*maoc-1* plasmid. The *dhs-28* gene was inserted into the pACYCDuet-1 vector at the EcoRI/NotI sites (resulting in an N-terminal His tag) to generate pACYCDuet-1-*dhs-28Δscp-2*. This construct lacks the *dhs-28* sequence encoding the SCP-2 (Sterol Carrier Protein-2) domain because of its interference with protein expression and lack of relevance for enzymatic activity. The *daf-22* gene was inserted into a modified pET-16b vector at the NcoI/NotI sites (resulting in a C-terminal His tag) to generate pET-16b-*daf-22* plasmid. MAOC-1, DHS-28ΔSCP-2, and DAF-22 were expressed and purified using a similar method as that used for ACOX-3, except that expression was induced with 0.8 mM IPTG and FAD was not included in protein lysis and purification buffer.

## Acyl-CoA oxidase activity assays

For the LC-MS-based assay, 40 µM IC-asc-C9-CoA, 20 µM FAD, 20 µM $NAD^+$, 200 µM CoA, 8 µg of ACOX protein (ACOX-1.1, ACOX-1.2, or ACOX-3), and 4 µg of other three β-oxidation enzymes (MAOC-1, DHS-28, and DAF-22) were added to the reaction buffer (100 mM Tris, 8 mM $MgCl_2$, pH 7.5), for a total reaction volume of 50 µL. The control reaction contained all of the above except the ACOX proteins. Reactions were performed at 30°C for 1 hr. Then 50 µL of MeOH was added to quench the reaction. Samples were heated at 95°C for 5 min and centrifuged at 13000 rpm for 5 min. 10 µL of the supernatant was used for LC-MS analysis. Retention times of the substrate and products were confirmed with synthetic standards IC-asc-C9, IC-asc-C7, IC-asc-C5, IC-asc-C9-CoA, and IC-asc-C7-CoA. The retention time of IC-asc-C5-CoA was predicted using the linear relationship between the retention times of IC-asc-C5-CoA and IC-asc-C7-CoA and IC-asc-C9-CoA. The coupled enzyme assay for acyl-CoA oxidase activity was performed as described previously (*Zhang et al., 2015*), with several modifications. Specifically, reactions were performed at room temperature (~23°C), and the substrate concentration was 24 µM. Three independent experiments were performed using protein purified at three different times.

## ACOX-1.1, ACOX-3, and ACS-7 localization and RNAi

To generate *Pacs-7::gfp::acs-7*, 2.2 kb of the *acs-7* promoter and the *acs-7* gene plus 3'-UTR were inserted into the AscI/NotI and NgoMIV/AatII sites, respectively, of pPD114.108 (from Andy Fire, via Addgene). 120 ng/µL of *Pacs-7::gfp::acs-7* was injected into wild-type worms to give RAB37 *rebEx11* (*Pacs-7::gfp::acs-7*). To generate *Pacox-1.1::mcherry::acox-1.1*, 1.4 kb of the *acox-1.1* promoter and the *acox-1.1* gene plus 3'-UTR were inserted into the SalI/NotI and NgoMIV/ApaI sites, respectively, of pPD114.108. The GFP sequence in the vector was replaced with the mCherry sequence amplified from pMC10-mCherry (gift of Piali Sengupta) using the AgeI/NheI sites. To generate *Pacox-3::gfp::acox-3*, 3 kb of the *acox-3* promoter and the *acox-3* gene plus 3'-UTR were inserted into the SalI/NotI and NheI/ApaI sites, respectively, of pPD114.108. 60 ng/µL of *Pacox-1.1::mcherry::acox-1.1* was co-injected with either 60 ng/µL of *Pacs-7::gfp::acs-7* or *Pacox-3::gfp::acox-3* into wild-type worms to give RAB38 *rebEx12* (*Pacox-1.1::mcherry::acox-1.1; Pacs-7::gfp::acs-7*) and RAB39 *rebEx13* (*Pacox-1.1::mcherry::acox-1.1; Pacox-3::gfp::acox-3*), respectively. Alternatively, 10 ng/µL of *Pacox-1.1::mcherry::acox-1.1* was co-injected with 120 ng/µL of *Pacs-7::gfp::acs-7* into wild-type worms to

give RAB40 *rebEx14* (*Pacox-1.1::mcherry::acox-1.1; Pacs-7::gfp::acs-7*). Imaging was conducted on a Zeiss Axiovert.A1 microscope equipped with ZEN lite 2012 camera. In the RNAi feeding assay, *E. coli* strain HT115 (DE3) carrying L4440 (control) or L4440-*prx-13* (from Julie Ahringer RNAi strain library) was cultured in LB medium (with 150 µg/mL ampicillin) at 37°C at 225 rpm for 5 hr followed by induction with 4 mM IPTG for 1 hr. 200 µL of the bacterial cultures were then seeded onto individual NGM agar plates (with 1 mM IPTG, 25 µg/mL carbenicillin) and the plates were dried overnight. On the next day, two *Pacs-7::gfp::acs-7; Pacox-1.1::mcherry::acox-1.1* worms at the L4 stage were transferred onto each plate. Adults were removed from the plates 2 days later (on day 3), and on day 5, the progeny were imaged. To make LysoTracker Red plates, 6 µL of the LysoTracker Deep Red (Thermo L12492, 1 mM stock in DMSO) were added to each 3 cm plate (containing 3 mL NGM-agar per plate) to obtain a final concentration of 2 µM. After the agar plates solidified, 50 µL of OP50 (freshly inoculated culture in LB medium, 37°C, 225 rpm, 6 hr) were added to the center of each plate. Plates with bacteria were dried in a sterile hood in the dark for about 20 min. Twenty RAB37 *rebEx11* (*Pacs-7::gfp::acs-7*) worms or wild-type worms were transferred to each plate while minimizing exposure to light under the microscope. The plates were kept in the dark for 24 hr before imaging.

## Acknowledgements

We thank Patrick McGrath and Wen Xu for kindly providing plasmids and advice for making mutations via CRISPR-Cas9. We also thank Piali Sengupta for plasmids. We acknowledge the kind gift of the *M. tuberculosis* genomic library from Yousong Ding and Peilan Zhang. We appreciate the advice of Justin Ragains and Kyle Hollister regarding the synthesis of the IC-ascarosides. We thank Nicole Horenstein for advice on the coupled enzyme assays and for the use of her UV spectrometer. We are grateful to Tim Garrett and the Southeast Center for Integrated Metabolomics for access to the triple-quadrupole mass spectrometer. We thank Kari Basso and Manasi Kamat for providing HR-LC-MS/MS analysis at the University of Florida Mass Spectrometry Research and Education Center, which is funded by the NIH (S10 OD021758-01A1). Some strains were provided by Shohei Mitani at the National Bioresource Project (Japan) and by the Caenorhabditis Genetics Center, which is funded by NIH Office of Research Infrastructure Programs (P40 OD010440). This work was supported by the NIH (GM118775), the NSF (Career, 1555050), the Ellison Medical Foundation, the Research Corporation for Science Advancement, and the Alfred P. Sloan Foundation.

## Additional information

### Funding

| Funder | Grant reference number | Author |
| --- | --- | --- |
| National Institutes of Health | R01 GM118775 | Rebecca A Butcher |
| National Science Foundation | CAREER Award - 1555050 | Rebecca A Butcher |
| Lawrence Ellison Foundation | New Scholar in Aging Award - AG-NS-0963-12 | Rebecca A Butcher |
| Research Corporation for Scientific Advancement | Cottrell Scholar Award - 22844 | Rebecca A Butcher |
| Alfred P. Sloan Foundation | Sloan Fellowship - BR2014-071 | Rebecca A Butcher |

The funders had no role in study design, data collection and interpretation, or the decision to submit the work for publication.

### Author contributions

Yue Zhou, Yuting Wang, Conceptualization, Resources, Data curation, Formal analysis, Validation, Visualization, Methodology, Writing—review and editing; Xinxing Zhang, Conceptualization, Resources, Data curation, Formal analysis, Validation, Methodology, Writing—review and editing; Subhradeep Bhar, Resources, Investigation, Visualization; Rachel A Jones Lipinski, Jungsoo Han, Resources;

Likui Feng, Conceptualization; Rebecca A Butcher, Conceptualization, Formal analysis, Supervision, Funding acquisition, Validation, Investigation, Visualization, Methodology, Writing—original draft, Project administration, Writing—review and editing

**Author ORCIDs**
Rebecca A Butcher 🔟 http://orcid.org/0000-0002-0925-4459

**Decision letter and Author response**
Decision letter https://doi.org/10.7554/eLife.33286.015
Author response https://doi.org/10.7554/eLife.33286.016

## Additional files

**Supplementary files**
• Transparent reporting form
DOI: https://doi.org/10.7554/eLife.33286.013

**Data availability**

All data generated or analysed during this study are included in the manuscript and supporting files.

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
