## [Decision Letter]

Thank you for submitting your article "Biosynthesis of Indole-Modified Ascaroside Pheromones in *C. elegans*" for consideration by *eLife*. Your article has been favorably evaluated by Michael Marletta (Senior Editor) and three reviewers, one of whom, Jon Clardy (Reviewer #1), is a member of our Board of Reviewing Editors.

The reviewers have discussed the reviews with one another and the Reviewing Editor has drafted this decision to help you prepare a revised submission.

Summary:

Small molecules called ascarosides govern many of the developmental and behavioral aspects of the important model organism *C. elegans*. Apparently small changes in the structure of an ascaroside can have profound effects on worm behavior. Shortening the chain length of a fatty acid tail by a few carbons can change behavior from the arrested metabolism of the persistent dauer state to aggregative behavior. The authors look at how the biosynthesis of one class of ascaroside, the IC-ascarosides, connects with the altered biological function. They correct a previous model in which the switch would require de novo synthesis of new IC ascarosides into a model where β-oxidation, a common metabolic reaction, changes the structure and function of an existing ascaroside.

Essential revisions:

Both reviewers found the new model of IC-ascaroside biosynthesis in which the short chain variants come from β-oxidation of the side chain to be convincingly documented. It also clarifies some puzzling features of the earlier mechanistic proposal and alters our understanding of how ascaroside signaling can be modified to respond to changing circumstances. There are only a few relatively minor comments on the technical aspects of the mechanistic analysis.

Both reviewers felt that, as written, the manuscript did not put forward a clear case for the updated mechanism's broader significance – a case they both agreed could have been easily made. The new mechanism allows an IC-ascaroside's function to be altered without the de novo synthesis of new ascarosides, and the altered length of the side chain dramatically alters the IC-ascarosides function. Is this shortening by β-oxidation and altered function connected to nutritional status, for example? Is the β-oxidation shortening model applicable to other ascarosides? These issues are hinted at in an unclear fashion in the Discussion at the very end of the manuscript, but they don't appear in the Introduction, which is not clearly written, or the Abstract. The typical *eLife* reader is unlikely to keep reading about mechanistic details without some promise that issues of greater significance will be addressed.

Title:

The current title doesn't reflect the full significance of the study that comes from linking biosynthesis to development/behavior. Something like "Biosynthetic tailoring of existing ascarosides alters their biological function."

---

## [Author Response]

Essential revisions:Both reviewers found the new model of IC-ascaroside biosynthesis in which the short chain variants come from β-oxidation of the side chain to be convincingly documented. It also clarifies some puzzling features of the earlier mechanistic proposal and alters our understanding of how ascaroside signaling can be modified to respond to changing circumstances. There are only a few relatively minor comments on the technical aspects of the mechanistic analysis.Both reviewers felt that, as written, the manuscript did not put forward a clear case for the updated mechanism's broader significance – a case they both agreed could have been easily made. The new mechanism allows an IC-ascaroside's function to be altered without the de novo synthesis of new ascarosides, and the altered length of the side chain dramatically alters the IC-ascarosides function. Is this shortening by β-oxidation and altered function connected to nutritional status, for example? Is the β-oxidation shortening model applicable to other ascarosides? These issues are hinted at in an unclear fashion in the Discussion at the very end of the manuscript, but they don't appear in the Introduction, which is not clearly written, or the Abstract. The typical eLife reader is unlikely to keep reading about mechanistic details without some promise that issues of greater significance will be addressed.

We have extensively revised the text of the manuscript. The Abstract, Introduction, and Discussion have been completely rewritten. In the revised text, we have emphasized that we have identified a mechanism that enables *C. elegans* to dynamically tailor the IC-ascarosides that it has produced. This tailoring enables the worm to respond rapidly to changing environmental conditions and alter its chemical message without having to synthesize new ascarosides de novo. The mechanism explains how *C. elegans* can shorten the side chains of medium-chain IC-ascarosides in response to conditions such as starvation in order to convert a favorable signal that induces aggregation to an unfavorable one that induces the stress-resistant dauer larval stage. This mechanism is also likely used by the worm to tailor the side chains of OS-modified ascarosides under starvation conditions in order to produce a dispersal signal.

Title:The current title doesn't reflect the full significance of the study that comes from linking biosynthesis to development/behavior. Something like "Biosynthetic tailoring of existing ascarosides alters their biological function."

We have changed the title to the suggested title.

Additionally, we have revised Figure 1 in order to clarify the differences between our model and the previous model for IC-ascaroside biosynthesis.